# Evaluation of Novel Embroidered Textile-Electrodes Made from Hybrid Polyamide Conductive Threads for Surface EMG Sensing

**DOI:** 10.3390/s23094397

**Published:** 2023-04-29

**Authors:** Bulcha Belay Etana, Benny Malengier, Timothy Kwa, Janarthanan Krishnamoorthy, Lieva Van Langenhove

**Affiliations:** 1Department of Materials, Textiles and Chemical Engineering, Ghent University, 9000 Gent, Belgium; 2Jimma Institute of Technology (JiT), School of Materials Science and Engineering, Jimma University, Jimma P.O. Box 378, Ethiopia; 3Medtronic, 710 Medtronic Parkway Minneapolis, Minneapolis, MN 55432-5604, USA; 4Jimma Institute of Technology (JiT), School of Biomedical Engineering, Jimma University, Jimma P.O. Box 378, Ethiopia

**Keywords:** textile sensor, embroidered electrodes, inter-electrode distance, bioelectrical impedance analysis, hybrid conductive threads, sEMG

## Abstract

Recently, there has been an increase in the number of reports on textile-based dry electrodes that can detect biopotentials without the need for electrolytic gels. However, these textile electrodes have a higher electrode skin interface impedance due to the improper contact between the skin and the electrode, diminishing the reliability and repeatability of the sensor. To facilitate improved skin–electrode contact, the effects of load and holding contact pressure were monitored for an embroidered textile electrode composed of multifilament hybrid thread for its application as a surface electromyography (sEMG) sensor. The effect of the textile’s inter-electrode distance and double layering of embroidery that increases the density of the conductive threads were studied. Electrodes embroidered onto an elastic strap were wrapped around the forearm with a hook and loop fastener and tested for their performance. Time domain features such as the Root Mean Square (RMS), Average Rectified Value (ARV), and Signal to Noise Ratio (SNR) were quantitatively monitored in relation to the contact pressure and load. Experiments were performed in triplicates, and the sEMG signal characteristics were observed for various loads (0, 2, 4, and 6 kg) and holding contact pressures (5, 10, and 20 mmHg). sEMG signals recorded with textile electrodes were comparable in amplitude to those recorded using typical Ag/AgCl electrodes (28.45 dB recorded), while the signal-to-noise ratios were, 11.77, 19.60, 19.91, and 20.93 dB for the different loads, and 21.33, 23.34, and 17.45 dB for different holding pressures. The signal quality increased as the elastic strap was tightened further, but a pressure higher than 20 mmHg is not recommended because of the discomfort experienced by the subjects during data collection.

## 1. Introduction

Monitoring the patient’s health condition at home has become crucial in modern society. The development of smart devices based on wearable medical technologies has improved healthcare accessibility and enabled real-time patient monitoring. Biological signals, such as electrocardiogram (ECG), electromyogram (EMG), and electroencephalogram (EEG), are required for medical diagnosis and therapeutic intervention [1]. Among these, wearable surface electromyography (sEMG) or wireless-based sEMG signal monitoring has been shown to detect the loss of muscle strength in patients [2]. Wearable technologies used in health monitoring systems such as wearable electronics and smart fabrics are flexible and embed different sensors and actuators into wearable devices, satisfying the needs of healthcare providers across domains. Wearable smart watches or smart clothes that are worn on the human body have better portability, compatibility, and interface with multiple devices [3,4,5]. 

Textile-based electrodes are created by knitting, weaving, and embroidering fabrics with conductive yarn or metal wire, or by coating or printing conductive material on a variety of substrate materials [6,7,8,9]. Pitou et al., (2020) studied the performance of hand-made or simple embroidery machine-made electrodes in detecting EMGs; besides their functionality, they have also pointed out their limitations in terms of aesthetics and mass production as smart clothes [10]. Goncu-Berk and Tuna (2021) examined the effectiveness of embroidered electrodes in detecting EMG signals and compared it with the traditional Ag/AgCl hydrogel electrodes. They created a prototype sleeve contrasted with raglan and set-in sleeves. According to their findings, a well-fit textile-based embroidery electrode when worn as a sleeve detected the EMG signal consistently [11]. This article provides a brief description of hybrid conductive threads and their specific applications in health monitoring systems. It also reports on the basic electrical and mechanical parameters required for the design of textile electronic elements. The authors also focused on the development of an EMG sensor made of conductive multifilament hybrid polyester threads (CleverTex) integrated into the textile substrates. In this study, we have used a hybrid thread from Clevertex with a linear resistance of ~110 Ω/m, containing conductive Cu/Ag polyamide fibers and a mass density of 6% copper, 94% polyamide with 50% Texcount. The long-term stability and resistance of such textile electrodes on routine maintenance (washing and drying) were evaluated [12].

An additional important parameter to be considered when developing such a textile electrode is the material’s sustainability, which significantly affects the quality of the sEMG signal. Ernest N. Kamavuako et al. developed embroidered EMG electrodes as an affordable alternative to the sensors used in prosthesis control by comparing their online and offline performance against conventional gel electrodes [13]. Nevertheless, motion artifacts remain one of the main obstacles that prevent its widespread use as a wearable monitoring system. To address this issue, the exertion of external pressure onto the electrode by inserting soft pads between the sensor and the textile have been reported by many researchers [14,15]. However, the optimum pressure to be applied was not easily deducible because of the variation in the conductivity of different materials and differences in the impedance characteristics of subjects. Jan Wang et al. used a non-contact textile electrode to mitigate allergic reactions on the skin and drying up of electrolytic gels. They developed a three-dimensional parametric model for motion artifacts by acquiring anthropometric data so as to control it. Furthermore, standardization techniques in the design of customized smart cloths were introduced to monitor ECG signals [15,16]. In addition to motion artifacts, EMG signals are also highly affected by physiological and anatomical characteristics of the muscles. As such, EMG represents the neuromuscular activities of the human body and require a different approach to overcome some of its inherent issues [17]. 

Here, we are interested in measuring the EMG signal using CleverTex hybrid threads with ultrafine metallic wires combined with synthetic filaments such as polyester or polyamide. Such threads are already being used in textile circuitry, sensors, or actuators, in healthcare, sport, automotive, workwear, and fashion. Embroidered circuitry is one of the few technologies that are applied to the garment after it has been manufactured, hence allowing the circuit geometry to be designed independently. In this article, we report the development of embroidered textile electrodes using the polyester multifilament conductive hybrid thread (CleverTex) with a resistance of 280 Ω/m [12], designed with the digital Brother PR670E embroidery machine (Figure 1). Due to its excellent mechanical qualities and the lack of loop creation during stitching, the CleverTex thread is ideally suited for embroidery. The thread could be used in a bobbin or needle and is suited for high production quality. The embroidery thread works flawlessly with the other commonly known cloth embroidery techniques. Since they do not cause skin irritation or toxicity in addition to washing sustainability, these threads are suitable for the design of smart wearable health monitoring devices. Additionally, silicone-paraffin emulsions are applied to these threads to improve their running properties during the embroidery process. With the advancements in fabrication technology, a variety of electrically conductive hybrid threads are realized by combining ultrafine metallic wires with synthetic filaments such as polyester or polyamide.

## 2. Materials and Methods

### 2.1. Materials

The polyester multifilament conductive hybrid thread (CleverTex) was used to fabricate the embroidered textile electrodes. To prepare the embroidery region, the polyester multifilament conductive hybrid thread was used as the needle thread and 100% polyester Madeira thread was used as the bobbin thread. An elastic, Nylon + Polyester (black color) based 8 mm bandage with size of 2.95 × 27.5 inches, manufactured by BXT group limited was attached on the adhesive paper to make an appropriate fixation under the embroidery machine. It exhibited self-adhesive fastener straps, felt comfortable, and had breathable mesh. The embroidered electrodes were designed using the Ink/Stitch software, an embroidery plugin for Inkscape. The design was uploaded into a computerized embroidery machine (Brother670E) to develop the electrodes, see Figure 2. The electrodes were fabricated with a stitch length of 1.5 mm.

### 2.2. Experimental Setup for Surface Resistance and Impedance

The embroidered electrodes were compared with standard electrodes made of gelled Ag/AgCl that comes with an adhesive patch (3MCompany, Maplewood, MN, USA). The surface resistance of embroidered electrodes was measured via the two-point probe method using a Fluke 87 multimeter. The two probes were placed at the opposite ends of the diameter (~20 mm) of the circular electrode’s patch area. The same sample was measured at different probing points to compute the mean and standard deviation (SD).

Ivium “CompactStat.h10800” connected with two embroidered electrodes on an elastic strap was used for impedance measurements, after wrapping it on the biceps muscle of the subject, see Figure 3. EMG signals were acquired using the BIOPAC Student Lab device (BIOPAC Systems, Inc., Goleta, CA, USA) with a three-electrode configuration. Data were visually inspected during acquisition and the digital signal processing, and feature extraction was performed using the MATLAB Simulink environment (MathWorks, Natick, MA, USA). Time domain features in the EMG signal such as the *RMS*, ARV, and *SNR* were determined as shown in Figure 4. The *SNR* (dB) for the effect of load, and pressure on embroidered EMG electrodes in comparison with gel electrodes. *SNR* (dB) was calculated using Equation (1).
(1)$$SNR\;\left(\mathrm{dB}\right)=20{\;\log}_{10}\left(\frac{RMS\left(\mathrm{signal}\right)}{RMS\left(\mathrm{Noise}\right)}\right)$$

### 2.3. EMG Recording Protocol

This study was approved by the Ethics Committee of Jimma University, Jimma Institute of Technology, Doctoral School for Materials Science and Engineering. The volunteers were informed about the nature of the study and briefed on the recording protocol in advance of signing a consent form. In total, three recording sessions were performed with 2 male and 1 female healthy subjects aged between 20 and 41 years, with body mass indices between 19.6 and 27.4 kg/m^2^. The recordings were carried out with the subjects seated. After placing the electrodes on the skin, the skin–electrode contact was allowed to stabilize for 5 min before recording the data from the biceps muscle [18]. The baseline EMG signals were collected with the subject placed in a seated position. The elbow was folded (flexion) until it reached 90° and then extended (extension) to 0°, to record the muscle contraction signals originating from the bicep muscles. A flexion and extension cycle took around 10 s and, in total, a minimum of 5 cycles were executed by each subject. The measurements were performed in triplicate for each experiment, see Figure 5. To avoid systematic errors, four levels (0, 2, 4, 6 Kg) of the load effect and four levels (0, 5, 10, 20 mmHg) of the pressure effect were carried out on 3 subjects in a random order. In the load experiment, the subjects were requested to lift different weights ranging between 0 and 6 Kg, in their palm/hand while the sEMG was being recorded.

### 2.4. Impedance Measurement Protocol

The dry EMG electrodes were connected to the CompactStat using connecting wires. Ivium software was used for acquiring and post-processing the impedance data. The measurements were performed in the frequency range of 0.1 Hz to 1 kHz, by applying alternating electrical current 0.025 A in accordance with the safety standards of the medical electrodes’ impedance measurements reported by Meiling Zhang et al. The AC impedance did not exceed 3 kΩ at 10 Hz and the input current did not exceed 100 µA, with the open-circuit voltage generally kept under 100 mV [19]. Two embroidered electrodes configuration was placed on a subject’s right forearm specifically biceps muscle at three different inter-electrode distances (IED) 15 mm, 20 mm, and 25 mm using a stretchable bandage and a contact pressure of 20 mmHg to guarantee stability and repeatability. Three measurements were recorded and averaged for each experiment to reduce the effect of power line interference noise contributed by measuring device and electrode movements.

### 2.5. Effect of Washing on Textile Electrode Performance

To investigate the wash stability and the embroidered electrodes’ resistivity performance, the embroidered electrodes were washed for up to twenty cycles in a domestic washing machine. As per the ISO norm, 105-C06-A1S, 2010 standards, the samples were washed with 4 g/L nonionic detergent and 1 g/L sodium carbonate at 40 °C for 30 min washing per cycle. In addition, the samples were dried with compressed air for 12 h at room temperature. To investigate the effects of washing on the electrical properties and wash stability performances of the embroidered electrode, the surface resistance of the washed electrodes was measured via the two-point probe method using a Falcondmm-10 multimeter across multiple participants as shown in Figure 6. Thereafter, the surface resistance increased almost four-fold. The mechanical action of the laundry affected the conductive truck of the samples and increased its electrical resistance.

### 2.6. Effect of Stretching on Textile Electrode Performance

The EMG samples were tested with 0%, 2%, 4% 6%, 8%, 10%, 12%, and 14% stretch of the base fabric. The strain is applied manually by rotating the rotor handle parts of the machine as shows in Figure 7. The relative resistance is calculated by Equation (2).
(2)$$R_{R}=\left(\frac{R_{f}}{R_{0}}\right)$$
where $R_{R}$ is relative resistance, ${\;R}_{f}$ denotes resistance after stretching, and $R_{0}$ the resistance before stretching. This experiment is used to determine the strain-dependent electrical resistance characteristic of the textile materials containing the EMG textile electrode by applying maximum or minimum stress/tension. 

## 3. Results

### 3.1. Electrode Impedance Characterization

The characteristics of the embroidered polyamide hybrid conductive thread-based dry EMG electrodes were investigated by measuring skin impedance without the use of conductive gel and skin preparations. The experimental setup is shown in Figure 5. The impedance characteristics of the developed electrode for different IEDs are shown in Figure 8a. It was observed that the inter-electrode distance (IED) of 25 mm provided the best comparable results with the gelled electrode, followed by 20 mm. Whereas, 15 mm IED showed higher impedance characteristics that negatively affected the quality of the acquired EMG signals. Since non-optimized inter-electrode distance can distort the target muscle signal and lead to incorrect interpretation of muscle activation, the inter-electrode distance is optimized prior to recording the sEMG signals using wearable sensors [20]. Even though the IED of 25 mm yielded a lower impedance value compared to 20 mm for all the frequencies studied, the correlation coefficient of 20 mm and 25 mm were comparable to that of the reference functional Ag/AgCl, whose IED was kept at 25 mm (refer Table 1). Considering the fact that the textile electrodes on usage could fold or shrink with a decrease in IED, we therefore chose to use the sub-optimal IED of 20 mm for further experiments. Further, it has been reported earlier that the lesser IED the lower the crosstalk contamination while recording sEMG experiments [20].

The effect of the double and single embroidered electrodes is shown in Figure 8b. The double embroidered electrodes showed lower impedance compared to the single layer, with improved signal characteristics. The increased conductive thread density is attributed to the enhancement in the performance of the electrode through reduced impedance [21]. To ensure the reproducibility of the recorded EMG signals, a reliability experiment was performed using three sets of similar electrodes. It is a well-known fact that textile electrodes show high variability when repeatedly measured on human skin. In earlier studies, such variations are explained through differences observed in the skin property of diverse subjects [22]. In this work, we used three identical electrodes, EM01, EM02, and EM03 made of the same material and dimension (20 mm in diameter) [23], see Figure 8c. In a reproducibility experiment, even though the standard Ag/AgCl electrodes with conductive gel showed a lower impedance compared to embroidered dry textile electrodes the correlation coefficients (i.e., covariance/standard deviation) of impedance among the triplicates were comparable and differed only marginally (refer Table 2).

### 3.2. EMG Signal Acquired Using Textile Electrodes

Initially, we measured the EMG data with different holding pressures of 5 mmHg, 10 mmHg, and 20 mmHg so as to optimize the signal with respect to the contact pressure as shown in Figure 9 [24]. Table 3 shows that the SNR value increased with pressure and reached an optimal value at 10 mM and started to decline at 20 mmHg. We identified that the maximum pressure of 20 mmHg used in the experiments caused more discomfort to the subjects during data collection. 

We also examined the effects of loads of 0 kg, 2 kg, 4 kg, and 6 kg on the EMG signals obtained using the embroidered textile electrodes made from polyamide conductive hybrid threads, as shown Figure 10. In the load effect experiments, we fixed the pressure and inter-electrode distances at 10 mmHg and 20 mm, respectively. With increased load, the time features of the EMG signal such as RMS, ARV, and SNR improved significantly, see Table 4. The results suggest that the load primarily decreased the contact impedance mediated by sweating of the skin during exercises. In fact, sweat is known to facilitate the flow of current at the skin–electrode interface and improve the signal quality [24,25].

The performance of the developed dry textile sensor was evaluated based on the average RMS and ARV values for the nine subjects who participated in the study as shown in Table 5. The results suggest that the intra and inter subject variation for these parameters is insignificant and the developed textile electrodes are considered to be suitable for recording high bio-signal quality required of medical devices.

The evaluation of RMS, and ARV of sEMG signals recorded by the designed textile electrodes before and after washes is listed in Table 6. After repeated washing, the electrode started to lose its initial conductivity. Though an increase in the absolute surface resistance was observed for repeated wash cycles, the electrodes still retained good electrical conductivity and time domain features, see Figure 11. The electrical properties of the designed textile electrodes were assessed through resistance measurements using a two point-probe methods after multiple wash cycles. The electrical resistance increased slightly after the first wash but was relatively stable until the fifth wash, Figure 12. Thereafter (in 6th wash cycle), the surface resistance increased four-fold. The mechanical action of laundry physically affect the conductive track of the samples leading to increased electrical resistance. The derived RMS, ARV and SNR values also indicated a marginal decrease for the Hybrid thread based-Embroidered electrode after five wash cycles, see Table 6.

The stretchability test studies the effects of tensile strain on electrical resistance of the EMG electrode as presented in Figure 13. The results of the stretchability test for the electrode show that the resistance of the developed electrode increased relatively by 0.25 times compared to its initial surface resistance, for an increase of 12% in the strain. The strain ($\varepsilon =\frac{\Delta L}{L0}$) is defined as the ratio of the change in length (ΔL) with respect to its initial length (L_0_), for the stretch or stress (σ) applied to the knitted elastic band. Beyond a threshold value for stress applied the strain experienced by the material remains constant and indicates its maximum stretchable length (ΔL_max_). When the base fabric is stretched, the cross sectional area (A) of the individual fiber decreases with a concomitant increase in its length (l) along the elongation axis. This leads to an increase in the conductive resistance ($R\propto \frac{l}{A}$) until a threshold strain value for the fabric is reached. Beyond this threshold value the stain of the fiber remains constant (i.e. its l,A), therefore the corresponding resistance also doesn’t vary. In practice, the wearable e-textile fabrics are not stretched significantly or subjected to high strains. Rather the fabrics will be subjected to repeated stretches of lower strains and hence its electrical resistance is bound to change less significantly over time. Our experiments show that the designed embroidered electrode is not affected significantly either by repeated washes or by strain resulting from stretches, indicating its suitability for sEMG measurements in biomedical applications. 

## 4. Discussion

The main challenge in implementing a textile dry electrode is the lack of an adhesive layer that firmly attaches the electrode to a confined area. Hence, an optimum pressure must be applied to enhance and maintain the skin–electrode contact for recording high-quality bio-potential signals [25]. It has been shown in [23]. that the SNR of the sEMG signal can be increased by optimizing the holding pressure of the cloth, primarily by decreasing the skin impedance. This optimized holding pressure range should also take into consideration the comfortability of the wearers. The contact impedance between the electrode and skin is relatively higher for dry electrodes compared to the traditional commercial electrodes because of the decreased sweat secretion and absence of the conductive electrode gel. Further, dry textile electrodes are prone to motion artifacts and are yet to be widely implemented in clinical settings. According to [14,26], when the pressure exerted on the electrode increases, the impedance decreases due to an increase in the contact area causing the flow of electrolytes to the electrodes with better SNR for signal [26]. Since the interface layer between the electrode and the skin is a thin layer of humidity, even small changes in the electrode location can cause large changes in the ionic distribution proximal to the electrode affecting its half-cell potential. The intrinsically hybrid thread-based embroidered textile electrode that is reported here shows excellent self-adhesiveness, stretchability, and conductivity properties. It exhibits similar skin-contact impedance and noise characteristics in comparison to the standard gel electrodes under static and dynamic measurements. We investigated the effect of pressure and force/load on skin–electrode impedance by applying pressures of 5 mmHg, 10 mmHg, and 20 mmHg that are within the optimal pressure range of the electrodes used in biomedical applications [14,27]. We found that the pressure had little effect on the impedance of the developed dry textile electrodes (Figure 8). Based on the better signal fidelity, a pressure of 10 mmHg was identified to be optimal for sEMG applications.

The amplitude of the sEMG signal is directly proportional to the degree/force and quantity of muscle fibers involved in the contraction process [28]. The progressive recruitment of larger and faster motor units (MUs) during muscle contraction, increases with the increase in the load (force)/exercises. The load improves the spectral parameters of the EMG signal [29]. During exercises, the bigger type II muscle fibers get activated first compared to the smaller type I muscle fibers. Since the propagation of conduction velocity (CV) depends on the diameter of muscle fibers [30], the type II fibers with larger diameters exhibit higher CV compared to type I fibers [29]. An increase in CV also implies an increase in the spectral parameters that characterize EMG. Despite many studies indicating improved EMG parameters for increased load, some studies that are based on power spectral density suggest that the observed spectral improvement is only up to a certain degree of force employed [29,31]. rather than for an extended range [21]. In this study, we observed that the ARV and SNR values increased monotonously in proportion to the loads which ranged between 0 and 6 kg. 

The embroidered electrodes exhibited acceptable RMS and ARV values even after multiple washing cycles with only a marginal increase in surface resistance (Table 6). In addition, the reliability test also indicated that the signal quality of the developed electrode is comparable to that of the traditional Ag/AgCl. The variation across subjects and reproducibility issues observed during the experiments can be explained through the differences in the acquisition times and non-identical environmental conditions, such as humidity and temperature. The sEMG-based functional tests performed here with reference to the standard electrode show the feasibility of the designed embroidered electrodes to record muscle activity and detect force and pressure variations. Wearable dry electrodes are much sought after in long-term biopotential monitoring but are limited by the effects of high interfacial impedance, motion artifacts caused by body movements and sweat secretions. The drawback we observed in developing the embroidered textile electrode is that the quality of the EMG signal decreased over time in comparison with the gel electrodes, exacerbated by motional artifacts. This problem can be addressed by optimizing both the holding pressure as well as moisture retention property of the textile electrode by wetting it with electrolytes. In the future, the moisture-retaining properties of textile electrodes will be explored by studying different moisture-absorbing fabric materials with better wetness properties independent of their environment.

## 5. Conclusions

Polyester multifilament conductive hybrid thread (CleverTex, Usti nad Orlice, Czech Republic) on cotton fabrics was used to design the textile electrodes for continuous EMG measurement. The measured surface resistance of the developed electrode was 1.34 Ω/sq. EMG signals recorded using the embroidered textile electrodes showed comparable RMS and ARV to that of the standard Ag/AgCl gel electrodes. The amplitudes of RMS, and ARV, were 0.40, and 0.26 mV for the embroidered textrode, respectively. The average SNR was 20.93, and 28.45 dB for the embroidered and gel electrode, respectively. Our results demonstrate that the EMG signal quality recorded using textile electrodes increased with an increase in the load and holding pressure. Only for signals collected at a 6 kg load and 10 mmHg holding contact pressure level, the SNR was comparable to that of the standard Ag/AgCl gel electrodes (Table 4). The developed textile electrodes also showed an acceptable range of SNR after five wash cycles despite a marginal increase in the surface resistance. In addition, this study demonstrates the potential feasibility of the use of embroidered electrodes for applications such as monitoring of muscle fatigue and prosthesis control [32,33]. Moreover, hybrid thread-based embroidered electrodes provide an affordable means to record sEMG for patients at home with minimal support from healthcare professionals. Future studies on diverse subject groups, over a long period of time will provide holistic view of the performance and degradation of hybrid thread-based electrodes with time. A comparison of sEMG signals from healthy volunteers and patients with an optimal inter-electrode distance for embroidered electrodes is in progress. 

## Figures and Tables

**Figure 1 sensors-23-04397-f001:**
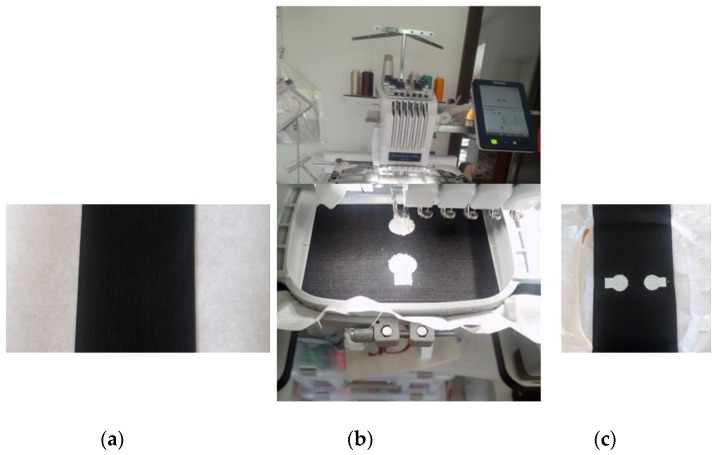
Experimental setup for Embroidery electrode fabrication: (**a**) Bandage attached on adhesive paper; (**b**) embroidering process; (**c**) embroidered electrode samples direct on bandage.

**Figure 2 sensors-23-04397-f002:**
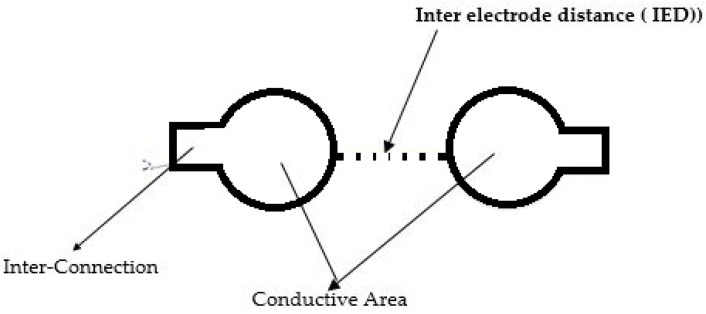
Designed embroidery electrodes using the Ink/Stitch software.

**Figure 3 sensors-23-04397-f003:**
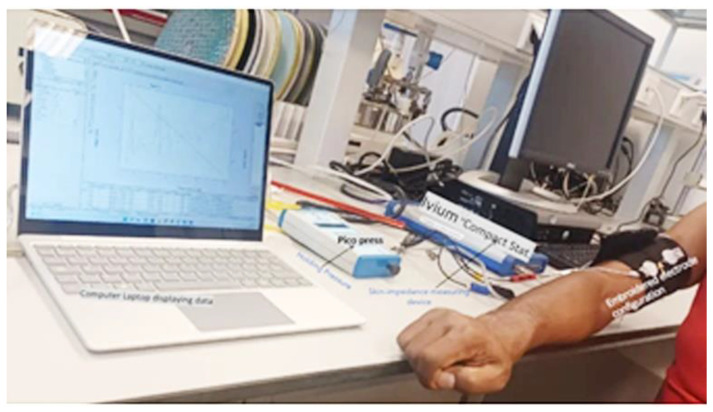
Experiment setup of textile EMG sensor with two-embroidered electrode configuration for impedance measurement.

**Figure 4 sensors-23-04397-f004:**
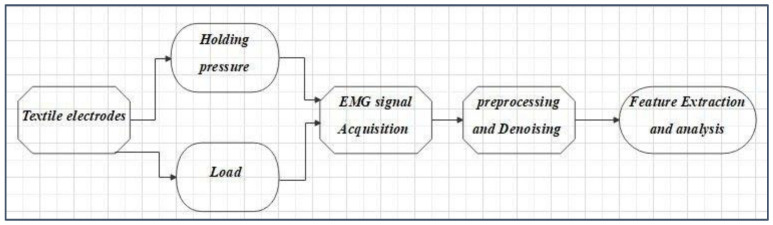
Block diagram of wearable textile electrode-based EMG sensor analysis.

**Figure 5 sensors-23-04397-f005:**
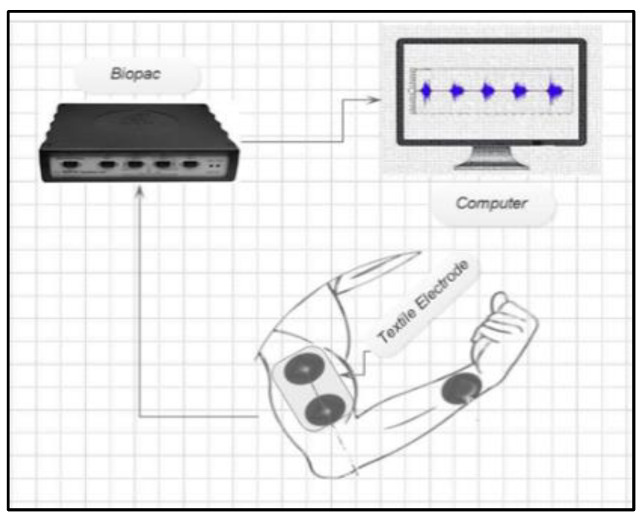
Schematic diagram of Experimental setup for EMG recording.

**Figure 6 sensors-23-04397-f006:**
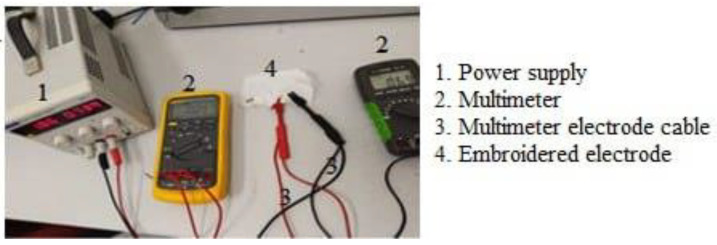
Surface electrical resistance measured using 2-point probe.

**Figure 7 sensors-23-04397-f007:**
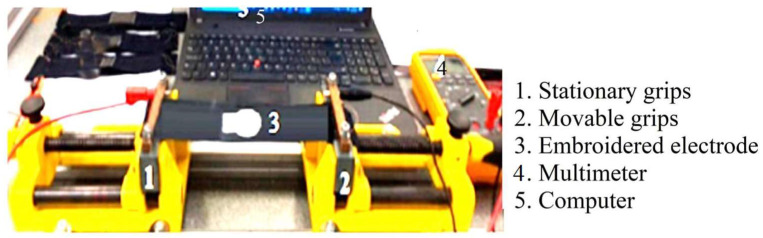
Measurement setups of effects of strain resistance of embroidered electrodes.

**Figure 8 sensors-23-04397-f008:**
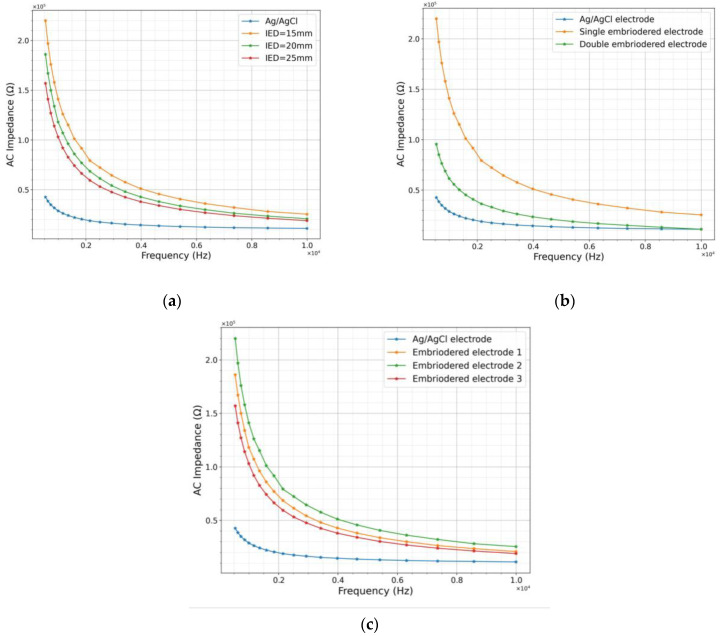
(**a**) Effect of inter-electrode distances (IED), (**b**) double layering, and (**c**) repetition with 3 similar samples (triplicates) on the impedance of the textile electrode in comparison with Ag/AgCl electrode measured on human biceps muscles.

**Figure 9 sensors-23-04397-f009:**
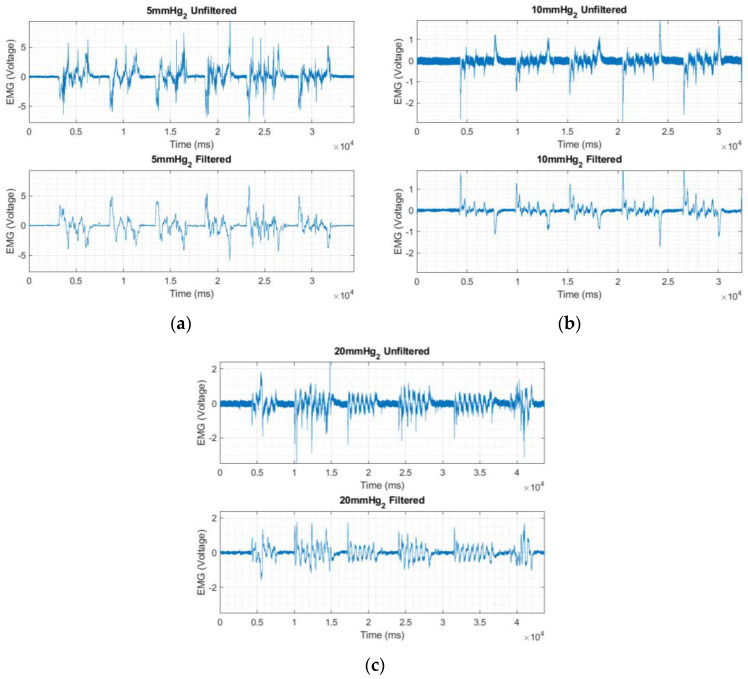
EMG signals collected using embroidered electrodes at 3 different holding pressures, (**a**) at 5 mm Hg, (**b**) at 10 mm Hg and (**c**) at 20 mm Hg.

**Figure 10 sensors-23-04397-f010:**
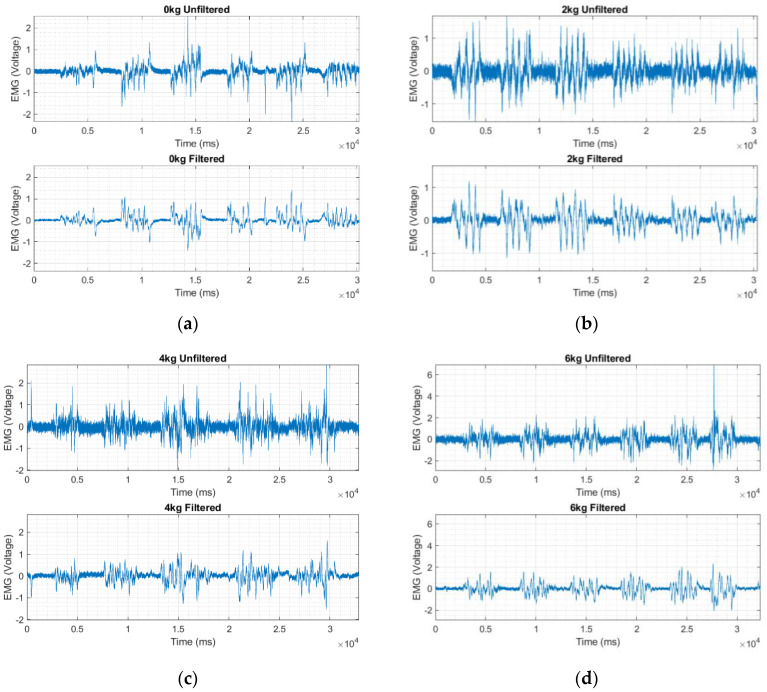
EMG signals collected using embroidered electrodes at different load levels, before and after filtering.

**Figure 11 sensors-23-04397-f011:**
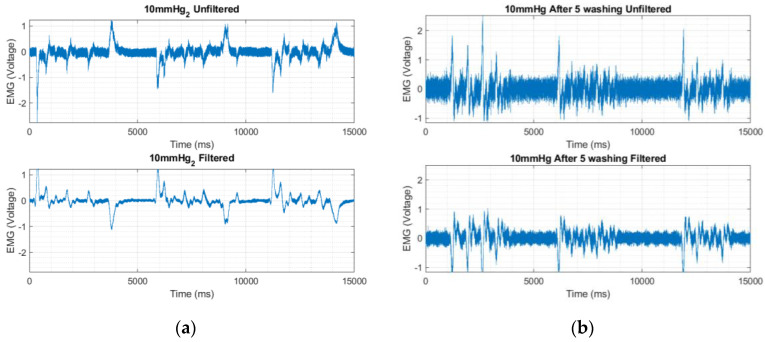
EMG signals recorded using hybrid thread-based embroidered electrodes: (**a**) before washing; and (**b**) after the fifth wash cycle out of the total six wash cycles.

**Figure 12 sensors-23-04397-f012:**
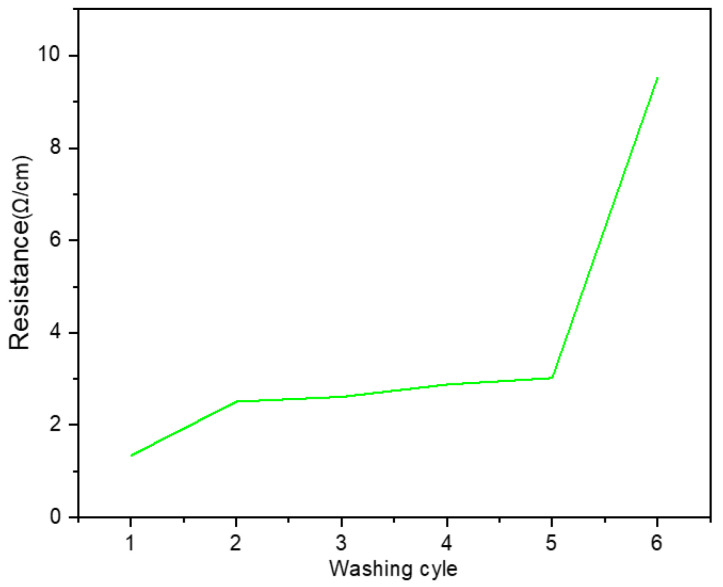
Change of surface resistance of hybrid thread-based embroidered textile electrodes with multiple washing cycles.

**Figure 13 sensors-23-04397-f013:**
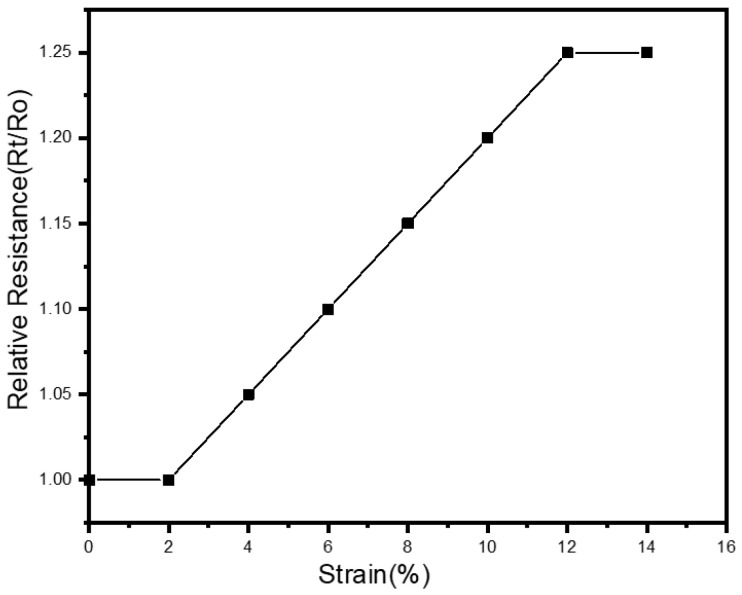
Effects of tensile strain on electrical resistance of EMG textile electrode.

**Table 1 sensors-23-04397-t001:** Correlation coefficients of impedance measured at different IEDs for embroidery electrodes. The *p* value for all the calculated coefficients were less than 10^−12^ at a significance of α = 0.05.

	Ag/AgCl	EM (15 mm)	EM (20 mm)	EM (25 mm)
Ag/AgCl (25 mm)	1.000	0.943	0.971	0.993
EM (15 mm)		1.000	0.943	0.953
EM (20 mm)			1.000	0.986
EM (25 mm)				1.000

**Table 2 sensors-23-04397-t002:** Correlation coefficients of impedance measured for three similar embroidery electrodes EM01, EM02, and EM03. The *p* values for all the calculated coefficients were less than 10^−12^ at a significance of α = 0.05.

	Ag/AgCl	EM01	EM02	EM03
Ag/AgCl (25 mm)	1.000	0.972	0.969	0.993
EM01		1.000	0.954	0.988
EM02			1.000	0.970
EM03				1.000

**Table 3 sensors-23-04397-t003:** Feature values of EMG signals collected using embroidered electrodes at different holding pressure without any load (i.e., 0 Kg).

Holding Contact Pressure (mmHg)	RMS (mV)	ARV (mV)	SNR (dB)
5	0.40 ± 0.02	0.16 ± 0.02	21.33 ± 1.13
10	0.47 ± 0.05	0.29 ± 0.04	23.34 ± 1.44
20	1.83 ± 0.1	1.06 ± 0.4	17.45 ± 1.43
Ag/AgCl (0 mmHg)	1.89 ± 0.05	1.07 ± 0.06	23.10 ± 1.33

**Table 4 sensors-23-04397-t004:** Feature values of EMG signals obtained for embroidered electrodes at different load levels.

Load (Kg)	Root Mean Square (RMS, mV)	Average Rectified Value (ARV, mV)	Signal-to-Noise Ratio (SNR, dB)
0	0.29 ± 0.05	0.18 ± 0.02	17.12 ± 1.74
2	0.30 ± 0.03	0.20 ± 0.04	19.53 ± 1.60
4	0.34 ± 0.05	0.22 ± 0.03	19.89 ± 1.91
6	0.55 ± 0.04	0.35 ± 0.05	20.18 ± 1.93
Ag/AgCl (0 Kg)	0.82 ± 0.02	0.62 ± 0.04	21.18 ± 1.72

**Table 5 sensors-23-04397-t005:** RMS and ARV values of EMG signal collected using embroidered textile electrodes for 9 subjects in triplicates were provided as mean ± SD. EMG was measured at an optimum holding pressure of 10 mmHg with 25 mm inter-electrode distance.

Subjects	RMS (mV)	ARV (mV)
1	0.08 ± 0.02	0.06 ± 0.01
2	0.14 ± 0.00	0.10 ± 0.00
3	0.15 ± 0.01	0.10 ± 0.00
4	0.106 ± 0.01	0.08 ± 0.01
5	0.13 ± 0.00	0.089 ± 0.00
6	0.010 ± 0.00	0.06 ± 0.00
7	0.16 ± 0.00	0.09 ± 0.00
8	0.13 ± 0.02	0.08 ± 0.01
9	0.12 ± 0.01	0.08 ± 0.01

**Table 6 sensors-23-04397-t006:** RMS, ARV, and SNR of the hybrid thread-based embroidered electrode before and after multiple washes. We observe a decrease in the RMS, ARV, and SNR of the hybrid thread-based embroidered electrode after wash cycles.

Parameters	Before Washing	After Washing
RMS (MV)	0.41 ± 0.04	0.31 ± 0.03
ARV (MV)	0.26 ± 0.02	0.18 ± 0.05
SNR dB	24.81 ± 1.19	21.00 ± 1.40

## Data Availability

Not applicable.

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
