# Peer review of "Evaluation of Novel Embroidered Textile-Electrodes Made from Hybrid Polyamide Conductive Threads for Surface EMG Sensing"

_sensors, 2023, doi:10.3390/s23094397_

Round 1

Reviewer 1 Report

This manuscript reported a multifilament hybrid thread-based embroidered textile electrode as a surface electromyography (sEMG) sensor and the effects of load and holding contact pressure were monitored by using the sensor. The manuscript could be accepted after some issues are properly addressed.

1: The formats of the whole manuscript need to be checked to make sure they are correct.

2: The quality of all figures should be improved

3: Some symbols are highlighted in red in line 171, please correct it.

4: In line 212, “the relative resistance is calculated by the equation 1” should be “… by the Equation (1)”

5: Errors can be found when citing references in the main text, such as that in line 298.

6: How many wash cycles was used in this work? It’s 5 washing cycles in Figure 11b, while 6 washing cycles in the corresponding caption.

7: The authors investigate the effect of loads of 0 kg, 2kg, 4kg, and 6 kg on the EMG signals. The author should provide the applied the loading area in order to calculate the pressure. Otherwise, it will be not rigorous.

8: In Figure 12, change resistance was used. However, I cannot find the distance between the two electrodes as shown in Figure 2. In addition, can the electrode shape affect the resistance measurement? The author should clarify it.

9: In figured 13, the relative resistance increased linearly with increasing the strain in the range of 2% and 12%. However, the relative resistance became stable when the strain was above 12%. The authors should explain the reason.

Author Response

Dear Editor,

We have addressed all the queries raised by both the reviewers and have amended the manuscript accordingly.

Reviewer 2 Report

Summary:

 The authors of this paper developed textile-based electrodes for potential use in wearable devices. The electrode is made of hybrid conductive threads by embroidery. The author evaluated the electrode configuration, i.e. the inter-electrode distance and the density of the conductive threads, and the external pressure in the influence of the skin-electrode impedance and the recorded sEMG signal quality. Despite the potential of the technology, the manuscript requires major revision before it can be considered for publication.

 Materials:

  1. More information on the CleverTex threads is needed, e.g. what are the conductive filaments? And how many filaments?  
  2. From line 116 is not ‘materials’ but testing setup (surface resistance measurement, impedance measurement).
  3. Line 133, reference load error
  4. From line 127- sEMG acquisition, the author wrote (line 133): the experiments were performed by flexing the elbow and measuring the contraction and relaxation of the bicep muscle. Please specify the contraction time, relaxation time and movement (contraction and relaxation) repetitions.
  5. 2.2 EMG recording protocol need to be restructured. Line 155-156, three consecutive trials for muscle contraction and relaxation, do you mean the 3 repeats of the two movements? And what do you mean by each phase lasted for 5 ms?
  6. Line 157-160, four conditions? What are the four conditions in both cases? This is unclear. And what do you mean in line 160 ‘(electrode sizes’)? Have you also studied electrode size?  
  7. Most of this part (from line 164) is not the ‘protocol’: line 164-176 is impedance measurement; from line 177 is background information; from 195 is about washability. From line 208 is strain-dependent resistance.  

 Methodology:

1. The authors used 2 points measurement for electrical resistance. Why not use 4-points, the more precise method?

2. sEMG at stretch (figure 7 and the related text): how would strain-dependent electrical resistance influence the EMG? I don’t understand why (surface)resistance before and after stretching is relevant.

3. Does sEMG depends on the IEDs? To my knowledge, there are no optimized IEDs in general, it depends on the muscles of the individual. Of course, different IDEs resulting different impedance, but in terms of sEMG, the common knowledge is ‘the small, the better’, and the standard IED for sEMG recording is 10mm.

Results and discussion:

  1. What is the IED of Ag/AgCl electrodes in the electrode impedance measurement setup? In figure 8, there is only one Ag/AgCl curve. Does it mean you only measured the impedance using Ag/AgCl electrode at one IDE? Or have you measured it at 3 different IDE and averaged the results?
  2. I think table 1 and 2 provide no useful information.
  3. Line 298, reference load error.
  4. Figure 9 , 10 and 11 need to be clarified. X-axis is ms x 10E+4, so it was recorded in 35 seconds . And you recorded 5/6 muscle contraction? This has to be reflected in either the method or the sEMG recording protocol part. And it has to be consistent.
  5. Figure 12, it is good to know the resistance change after washing cycles, but I think it is more interesting to know the sEMG recording (SNR) after washing.
  6. Same as in figure 13, sEMG after stretching is more interesting, although it is also not that relevant.
  7. Line 357, repeating the method

Conclusion:

Line 443, signals collected at 6kg load and 10mmHg… was comparable to the signals collected using Ag/AgCl electrode. How did you come up with this conclusion? I can't read any signal collected by Ag/AgCl electrode in the result section.

Author Response

(The authors gave the same response as above.)

Round 2

Reviewer 2 Report

All my comments are well addressed.